# Controlling network ensembles

Isaac Klickstein [1✉] & Francesco Sorrentino [1✉]

The field of optimal control typically requires the assumption of perfect knowledge of the system one desires to control, which is an unrealistic assumption for biological systems, or networks, typically affected by high levels of uncertainty. Here, we investigate the minimum energy control of network ensembles, which may take one of a number of possible realizations. We ensure the controller derived can perform the desired control with a tunable amount of accuracy and we study how the control energy and the overall control cost scale with the number of possible realizations. Our focus is in characterizing the solution of the optimal control problem in the limit in which the systems are drawn from a continuous distribution, and in particular, how to properly pose the weighting terms in the objective function. We verify the theory in three examples of interest: a unidirectional chain network with uncertain edge weights and self-loop weights, a network where each edge weight is drawn from a given distribution, and the Jacobian of the dynamics corresponding to the cell signaling network of autophagy in the presence of uncertain parameters.

---

[1] Department of Mechanical Engineering, University of New Mexico, Albuquerque, NM, USA. ✉email: iklick@protonmail.com; fsorrent@unm.edu

Our ability to numerically solve and implement optimal controls[1–3] has improved greatly this decade, but one typically must assume that nearly perfect knowledge of the system is available[4]. While this is usually not an issue for mechanical or designed systems[5], the optimal control of biological systems, or networks, cannot yet provide certain mathematical models[6]. There are several reasons why the underlying network structure and parameters may be affected by uncertainty: (i) our knowledge of the network connections may be imperfect, e.g., due to noisy measurements, (ii) networks change with time so a change may occur between the time the network is measured and the time when a control action is introduced and (iii) measurements performed by different research groups or by the same group under different environmental conditions may differ from each other. As an example of (iii), one can find several versions of the neural network of the worm *C. Elegans* in the literature[7,8] or the metabolic network of *E. Coli*[9,10], or variations between brain scans over time of the same individual[11]. The *E. Coli* protein–protein interaction network[12] is known to be affected by higher uncertainty than its metabolic network. Another example of (iii) is taking into account the effect of the environment on the transition rate between metabolites[13].

While considerable research efforts have been addressed at designing control laws for biological networks and other networked systems[14–21], a main limitation of these approaches is that an accurate mathematical model of these systems is typically unavailable. Recent work on applying optimal control to autophagy in cells[22] and regulating glucose levels in type 1 diabetes[23] required applying the resulting control to many possible realizations of the set of parameters to demonstrate their robustness. While the optimal control can be derived for any particular system realization, the resulting control is only optimal for that system. Instead, here, we derive the optimal control that is applicable to any one of a large number of possible realizations simultaneously. This has remained a fundamental open question; how can an optimal control be applied to systems and networks affected by uncertainty.

In general, uncertainty can appear in the form of both measurement and process noise affecting the system dynamics. The prototypical example of uncertainty entering a system is in the form of additive Gaussian noise, which in the case of a linear system and quadratic objective function, leads to the solution of the classical optimal control problem known as the linear-quadratic-Gaussian regulator[24]. In the field of stochastic optimal control[25], a control is derived for a system described by stochastic differential equations.

A classic approach for handling uncertainty is encompassed by the field of robust control, which is concerned with both model inaccuracies, such as imperfect knowledge of the system parameters, and the effects of exogenous disturbances, which naturally arise in any experimental setting. The problem of integrating optimal control and robustness has been dealt with in a number of classic works[26–32]. Feedback is known to significantly increase robustness of the control action[33]. The use of Lyapunov functions to design feedback controls for nonlinear systems with guaranteed optimality has also been addressed[34]. However, performing closed loop control of biological systems is notoriously difficult due to the lack of availability of real time measurements. Here we deal with a different problem for which open loop control is applied to an uncertain system and we are willing to accept an increase in cost to accommodate for such uncertainty. Instead of using common approaches such as system identification or learning, we study how the solution of the optimal control problem changes as uncertainty (i.e., the number of possible system realizations) grows and compute scaling relations for how the solution of the optimal control problem varies in response to increasing uncertainty. Our results are relevant to systems and networks, for which identification may not be viable, such as biological time-evolving systems.

The minimum energy control of complex networks has recently been used to analyze the controllability of complex networks[14,35,36] and our ability to allocate resources spatially to perform desired control tasks[37–40]. The work on controlling complex networks has currently centered around linear systems, which typically only provide rough approximations of biological systems as they normally exhibit multiple attractors. Nonetheless, examining linear systems has provided useful results[18] that can be used in experiments. Consider the general network ensemble described in Fig. 1, where the weight associated with each network edge is drawn from a given distribution. For example, for gene regulatory networks the weight distributions are typically estimated from a series of expensive measurements, performed in a noisy environment[41–43]. The main question we address in this paper is whether it is possible to design an optimal control strategy for a network ensemble, like the one presented in Fig. 1. By network ensemble, we mean a family of weighted, possibly directed, networks that satisfy a set of constraints[44–46], also sometimes called the canonical weighted network ensemble[47]. One possible solution to our proposed problem is to incorporate robustness in the optimal control strategy so that the strategy is effective regardless of the particular network realization drawn from the ensemble. Imagine for example to sample a number of network realizations $A^{(0)}, A^{(2)}, \ldots, A^{(N-1)}$ from the ensemble, such as those networks whose edge weights correspond to the distributions shown in Fig. 1. This problem is addressed by the optimal control problem discussed in the remainder of this paper, with particular focus on the case when $N \to \infty$, thus ensuring one can control a possibly infinite ensemble of systems.

To provide an example of a situation to which our analysis applies, imagine seeking to control a biological network for which certain connections are well known and characterized by experimental measures but other connections are not. For example, it may be uncertain whether given pairs of nodes are connected or not, and as a result, the network may exist in many different configurations based on which connections are present. Our approach described in the rest of this paper is based on

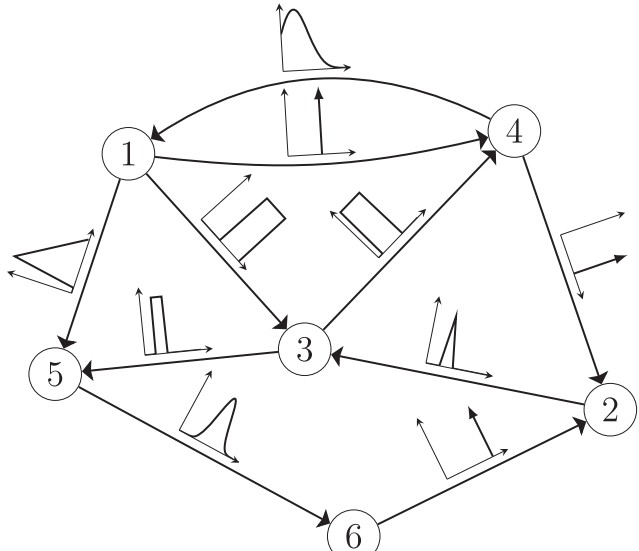

**Fig. 1 A network ensemble described by edge weights each drawn from a distribution.** A network with $n = 6$ nodes and $|\mathcal{E}| = 10$ directed edges. The edge weight associated with each edge is not known precisely but is instead drawn from some distribution indicated by the plots along each edge.

incorporating such uncertainty in the control action and in designing a control solution that works for all the possible network configurations. Similarly we can deal with other sources of uncertainty, e.g., affecting the weights associated with the network connections.

## Results

We consider systems which can be described by the triplet $(\mathcal{A}, B, C)$ where $\mathcal{A} = \{A_j \in \mathbb{R}^{n \times n} | j = 0, \ldots, N-1\}$ is a sample of $N$ square matrices describing a selection of networks from the network ensemble of interest, each of dimension $n$-by-$n$, the $n$-by-$m$ matrix $B$ which describes how the inputs are attached to the system and the $p$-by-$n$ matrix $C$ describes the relevant outputs of the system. Each matrix $A_j$ corresponds to a different version of the system one is trying to control. As the input and output matrices, $B$ and $C$, are often designed, we assume that they are known exactly, but extensions to the case where $B$ and $C$ are also drawn from a distribution, i.e., for each $A_j$ there is a corresponding $B_j$ and $C_j$ is straightforward. The time evolution of the states of this systems are described by the following set of $N$ systems of $n$ scalar linear differential equations.

$$\dot{\boldsymbol{x}}_j(t) = A_j \boldsymbol{x}_j(t) + B\boldsymbol{u}(t), \quad \boldsymbol{x}_j(0) = \boldsymbol{x}_0$$
$$\boldsymbol{y}_j(t) = C\boldsymbol{x}_j, \quad j = 1, \ldots, N \tag{1}$$

The ensemble of state matrices may be chosen as weighted adjacency matrices of graphs as shown in Fig. 1 or as the Jacobian of a nonlinear system where the parameters of the system are unknown. Both of these types of systems are investigated in the examples described later in this paper.

A small example of this type of composite system is shown in Fig. 2. Consider a five state linear dynamical system whose state matrix can be described by the adjacency matrix of a network shown on the top of Fig. 2 where the single control input is assigned to node 4 (in blue) so $B = \boldsymbol{e}_4$ and there is a single output,

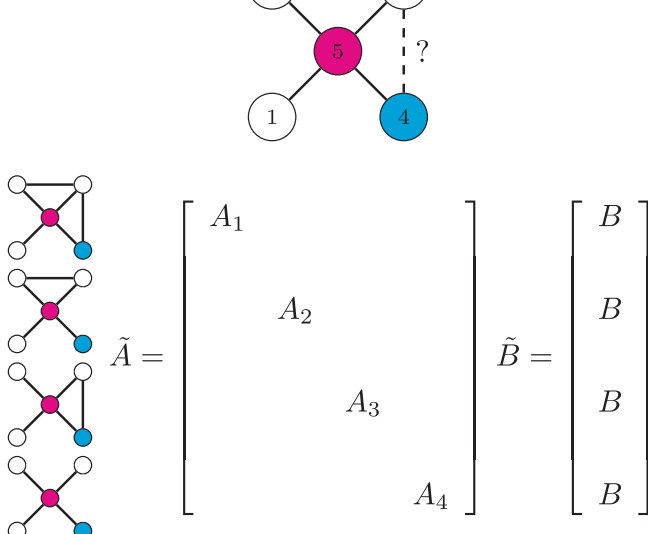

**Fig. 2 An outline of the method in terms of composite matrices $\tilde{A}$ and $\tilde{B}$.** A system that can be described as a network is shown at the top where the presence of two edges, (2, 3) and (3, 4) is uncertain. Driver nodes are in blue and target nodes are in magenta. Then make $N$ copies containing each possible network which contains a combination of those two edges. The composite adjacency matrix, $\tilde{A}$, is block diagonal with each corresponding network's adjacency matrix along the diagonal. The composite input matrix, $\tilde{B}$, consists of $N$ copies of $B$ stacked on top of each other.

node 5 (in magenta), so $C = \boldsymbol{e}_5^T$, where $\boldsymbol{e}_k$ is the $k$th unit vector. Two of the edges, drawn with a dash pattern, may or may not exist in the actual system. The $N = 4$ possible configurations are shown along the left hand side of Fig. 2, each of which can be represented by an adjacency matrix $A_k$, $k = 1, \ldots, 4$. The composite adjacency matrix of all possible configurations, denoted $\tilde{A}$, is a block diagonal matrix with each adjacency matrix, $A_k$, $k = 1, \ldots, 4$, assigned along its diagonal. The composite input matrix, denoted $\tilde{B}$, consists of $N$ copies of the input matrix $B$ stacked on top of each other. Similarly, the composite output matrix, denoted $\tilde{C} \in \mathbb{R}^{Np \times Nn}$, consists of $N$ copies of the output matrix $C$, placed diagonally in the block diagonal matrix. Thus, the original system written in Eq. (1) can equivalently be written in terms of the composite system $\dot{\boldsymbol{x}}(t) = \tilde{A}\boldsymbol{x}(t) + \tilde{B}\boldsymbol{u}(t)$ and $\boldsymbol{y}(t) = \tilde{C}\boldsymbol{x}(t)$ where $\boldsymbol{x}(t) = \left[\boldsymbol{x}_0^T(t) \cdots \boldsymbol{x}_{N-1}^T(t)\right]^T$.

The control energy (or effort) of the control input is defined as,

$$E = \int_0^{t_f} ||\boldsymbol{u}(t)||_2^2 dt \tag{2}$$

while the deviation of the control action is defined as,

$$D = \sum_{j=0}^{N-1} ||\boldsymbol{y}_j(t_f) - \boldsymbol{y}_f||_2^2 \tag{3}$$

where $\boldsymbol{y}_f \in \mathbb{R}^p$ is some desired final output of the system regardless of the realization. Note that the accuracy is a variance-like term if $\boldsymbol{y}_f$ is the average final state over the $N$ possible systems. We would like to design an optimal controller which is able to balance the control energy in Eq. (2) and the accuracy in Eq. (3)[48] of the control action,

$$\begin{aligned} \min \quad & J = \frac{(1-\alpha)}{2}D + \frac{\alpha}{2}E, \quad \alpha \in (0,1) \\ \text{s.t.} \quad & \dot{\boldsymbol{x}}_j = A_j \boldsymbol{x}_j(t) + B\boldsymbol{u}(t), \quad j = 1, \ldots, N, \\ & \boldsymbol{y}_j(t) = C\boldsymbol{x}_j(t), \\ & \boldsymbol{x}_j(0) = \boldsymbol{x}_0, \end{aligned} \tag{4}$$

The optimal control problem in Eq. (4) is solved using Pontryagin's Minimum Principle, for which the details are shown in Supplementary Note 1.1. Before presenting the solution, a few values must be defined. The variable $\alpha$ $(1 - \alpha)$ in Eq. (4) measures the relative weight assigned to the control energy (the deviation) in the objective function. The solution of the minimum energy control problem, that is $\min J = E$ with assigned terminal constraints $\boldsymbol{y}_j(t_f) = \boldsymbol{y}_f$, is recovered in the limit $\alpha \to 0$[48]. On the other hand, setting $\alpha = 1$ corresponds to assigning a zero cost to the deviation $D$, for which trivially the solution is $\boldsymbol{u}(t) = \boldsymbol{0}$. This case is of no interest and thus is not considered here. The matrix that plays the central role in all of the following results is the $Np$-by-$Np$ symmetric positive semi-definite matrix we call the composite output controllability Gramian (COCG),

$$\overline{W}(t) = \begin{bmatrix} CW_{1,1}(t)C^T & CW_{1,2}(t)C^T & \cdots & CW_{1,N}(t)C^T \\ CW_{2,1}(t)C^T & CW_{2,2}(t)C^T & \cdots & CW_{2,N}(t)C^T \\ \vdots & \vdots & \ddots & \vdots \\ CW_{N,1}(t)C^T & CW_{N,2}(t)C^T & \cdots & CW_{N,N}(t)C^T \end{bmatrix} \tag{5}$$

where the square matrices $W_{j,k}(t_f) \in \mathbb{R}^{n \times n}$ are the solutions of the differential Sylvester equation,

$$\dot{W}_{j,k}(t) = A_j W_{j,k}(t) + W_{j,k}(t)A_k^T + BB^T$$
$$W_{j,k}(0) = O_n, \quad j, k = 1, \ldots, N \tag{6}$$

evaluated at time $t = t_f$. For $k = j$, the solution of Eq. (6) is the

output controllability Gramian for each individual network configuration. The vector $\boldsymbol{\beta}_j = Ce^{A_jt}\boldsymbol{x}_0 - \boldsymbol{y}_f, j = 0, \ldots, N-1$ is the control maneuver of the $j$th system and $\boldsymbol{\beta} = (\boldsymbol{\beta}_0^T, \ldots, \boldsymbol{\beta}_{N-1}^T)^T$ collects all of the control maneuvers and $\boldsymbol{\gamma}_j = C\boldsymbol{x}(t_f) - \boldsymbol{y}_f, j = 0, \ldots, N-1$ is the accuracy of the $j$th system and $\boldsymbol{\gamma} = (\boldsymbol{\gamma}_0^T, \ldots, \boldsymbol{\gamma}_{N-1}^T)^T$ collects all of the accuracy vectors. To find the unknown accuracy vector $\boldsymbol{\gamma}$, we solve the following linear system of equations,

$$\left(\alpha I_{Np} + (1-\alpha)\bar{W}(t_f)\right)\boldsymbol{\gamma} = \bar{U}(\alpha)\boldsymbol{\gamma} = \alpha\boldsymbol{\beta}. \tag{7}$$

With the solution of this linear system, the total cost, the control energy, and the deviation can be expressed as quadratic forms (details are contained in Supplementary Note 1.2).

$$
\begin{aligned}
J_N(\alpha) &= \frac{\alpha(1-\alpha)}{2}\boldsymbol{\beta}^T(t_f)\bar{U}^{-1}(\alpha)\boldsymbol{\beta}(t_f) \\
E_N(\alpha) &= (1-\alpha)^2\boldsymbol{\beta}^T(t_f)\bar{U}^{-1}(\alpha)\bar{W}(t_f)\bar{U}^{-1}(\alpha)\boldsymbol{\beta}(t_f) \\
D_N(\alpha) &= \alpha^2\boldsymbol{\beta}^T(t_f)\bar{U}^{-1}(\alpha)\bar{U}^{-1}(\alpha)\boldsymbol{\beta}(t_f)
\end{aligned}
\tag{8}
$$

Let the eigendecomposition of the composite output controllability Gramian $\bar{W}(t_f) = \Xi\mathcal{M}\Xi^T$ where the columns of $\Xi$, $\boldsymbol{\xi}_k$, are the orthogonal eigenvectors and the diagonal entries of $\mathcal{M}$, $\mu_k$, are the eigenvalues of $\bar{W}(t_f)$. We order the eigenvalues in descending order, that is, $\mu_k \geq \mu_{k+1}$. Note that $\bar{U}(\alpha)$ and $\bar{W}(t_f)$ are simultaneously diagonalizable which means they share their eigenvectors, but for each eigenvalue of $\bar{W}(t_f)$, $\mu_k$, there is a corresponding eigenvalue of $\bar{U}(\alpha)$ denoted $v_k = (\alpha + (1-\alpha)\mu_k)$. The optimal cost, control energy and deviation can equivalently be written as summations in terms of the eigenvalues of $\bar{W}(t_f)$ defining $\theta_k = \boldsymbol{\beta}^T\boldsymbol{\xi}_k$ as described in Supplementary Note 1.3,

$$
\begin{aligned}
J_N(\alpha) &= \frac{\alpha(1-\alpha)}{2}\sum_{k=0}^{Np-1}\frac{\theta_k^2}{\alpha + (1-\alpha)\mu_k} \\
E_N(\alpha) &= (1-\alpha)^2\sum_{k=0}^{Np-1}\frac{\theta_k^2\mu_k}{(\alpha + (1-\alpha)\mu_k)^2} \\
D_N(\alpha) &= \alpha^2\sum_{k=0}^{Np-1}\frac{\theta_k^2}{(\alpha + (1-\alpha)\mu_k)^2}
\end{aligned}
\tag{9}
$$

respectively. The behaviors of the cost, control energy, and accuracy in Eq. (9) depend on (i) the projection of the control maneuver on each of the eigenvectors, $\theta_k$, (ii) their corresponding eigenvalues, $\mu_k$, as well as (iii) the particular choice of relative weight $\alpha$.

To determine the behavior of the cost, the control energy, and the deviation, as expressed in Eq. (9) as a function of $N$, we make the following two assumptions:

**Assumption 1** : $\quad \mu_k \approx \mu_0 r_1^k, \qquad\qquad \mu_0 \approx c_1 Np$

**Assumption 2** : $\quad \theta_k^2 \approx \max\{\theta_0^2 r_2^k, \theta_c^2\}, \quad \theta_0^2 \approx c_2 Np$

The quantities $r_1$, $r_2$, $c_1$, $c_2$, and $\theta_c^2$ are assumed to be, for large enough $N$, invariant with respect to the underlying distribution from which the matrices $A^{(j)}$ are drawn. Note that the values $\theta_k^2$, $k = 1, \ldots, N$, depend on the final state, $\boldsymbol{y}_f$, and so by choosing a different $\boldsymbol{y}_f$, the specific values of $\theta_k^2$ will also change. Nonetheless, we have seen that while the specific choice of $\boldsymbol{y}_f$ affects the small fluctuations around the approximation, for $N \gg p$, the exponential scaling still holds. For all network ensembles examined by the authors these assumptions have held true, and their numerical calculations are presented alongside the results contained in this paper.

In the following section, we present our main result, that under the proper choice of $\alpha = \alpha(N)$, as $N \to \infty$, the control energy

$E_N(\alpha)$ is upper bounded by a constant value (as are the average deviation $D_N(\alpha)/N$ and the total cost $J_N(\alpha)$), as long as Assumption 1 and Assumption 2 hold. The main result of our work investigates the behavior of the control energy and the average deviation in Eq. (9) after applying Assumption 1 and 2, which we call $\bar{J}_N(\alpha)$ (the approximate total cost), $\bar{E}_N(\alpha)$ (the approximate control energy), and $\bar{D}_N(\alpha)/Np$ (the approximate average deviation) to make clear the dependence on both $\alpha$ and $N$ and the dependence on the assumptions listed above. In particular, we derive the appropriate values of $\alpha = \alpha(N)$ so that the control energy remains finite even in the $N \to \infty$ limit while bounding the average deviation below an arbitrarily small value.

In Supplementary Note 1.4, an upper and lower bound for $\bar{E}_N(\alpha)$ is derived, namely, $\bar{E}_{N,LB}(\alpha) < \bar{E}_N(\alpha) < \bar{E}_{N,UB}(\alpha)$. These bounds allow us to investigate the role of $\alpha$ in the $N \to \infty$ limit. A trivial choice of $\alpha$ is one that is independent of $N$. However, we show in Supplementary Note 1.5 that such a choice renders the solution of the optimal control problem infeasible in the large $N$ limit. Instead, we see that the control energy remains bounded in the $N \to \infty$ limit if and only if $\lim_{N\to\infty}\frac{(1-\alpha)Np}{\alpha} = b$. With this in mind, we choose the weighting parameter $\alpha = \alpha(N)$ to be,

$$\alpha(N) = \frac{Np}{Np+b}, \quad b > 0 \tag{10}$$

which maps the interval $\alpha \in (0, 1)$ to $b \in (0, \infty)$ where $b = 0$ corresponds to $\alpha = 1$ and $b \to \infty$ corresponds to $\alpha \to 0$. The values of $\bar{E}_N(\alpha)$, $\bar{D}_N(\alpha)/Np$, and $\bar{J}_N(\alpha)$ with the choice of $\alpha(N)$ in Eq. (10) are shown in Supplementary Note 1.4. In Supplementary Note 1.5, the approximations can all be shown to be upper bounded by the following expressions,

$$
\begin{aligned}
\bar{J}_N(b) &\leq b\frac{Np}{2(Np+b)}\left[c_2\frac{1-r_2^{\bar{k}+1}}{1-r_2} + \theta_c^2\right] \\
\bar{E}_N(b) &\leq b^2c_1\left[c_2\frac{1-(r_1r_2)^{\bar{k}+1}}{1-r_1r_2} + \frac{\theta_c^2}{Np}\frac{r_1^{\bar{k}+1}-r_1^{Np}}{1-r_1}\right] \\
\bar{D}_N(b)/Np &\leq \left[c_2\frac{1-r_2^{\bar{k}+1}}{1-r_2} + \theta_c^2\right]
\end{aligned}
\tag{11}
$$

From Eq. (11) we immediately see that for any value of $b$ the upper bounds on both $\bar{E}_N(b)$ and $\bar{J}_N(b)$ approach finite values in the limit $N \to \infty$. Moreover, two important facts follow from the inequalities in Eq. (11); (i) the upper bound of the control energy approximation grows quadratically in $b$ and (ii) the upper bound of the average deviation approximation is independent of $b$. In addition, by its definition, the average deviation is a strictly nonnegative value, $\bar{D}_N(b)/Np \geq 0$ for $b \in (0, \infty)$. The inequality only becomes an equality in the limit as $b \to \infty$. By taking the derivative of $\bar{D}_N(b)/Np$ with respect to $b$, the average deviation is shown to be monotonically decreasing in $b$ in Supplementary Note 1.5. Thus, there exists a value of $b$ such that the average deviation can be made smaller than any positive value $\epsilon$.

While the average deviation can be bounded below an arbitrarily small value for large enough $b$, it is important to note that the upper bound of the control energy grows quadratically in $b$, re-expressing the weight that appeared in the optimal control problem in Eq. (4) as a single parameter trade-off in $b$. This trade-off between accuracy and control energy is an important consideration for the scientist to consider when designing the controller. For example, if one is interested in reducing $\bar{D}_N(b)/Np$ and to do so requires multiplying $b$ by 10, then the upper bound of the control energy will increase by one hundred.

Through the following examples, the control energy, average deviation, and total cost are shown to be bounded as stated in

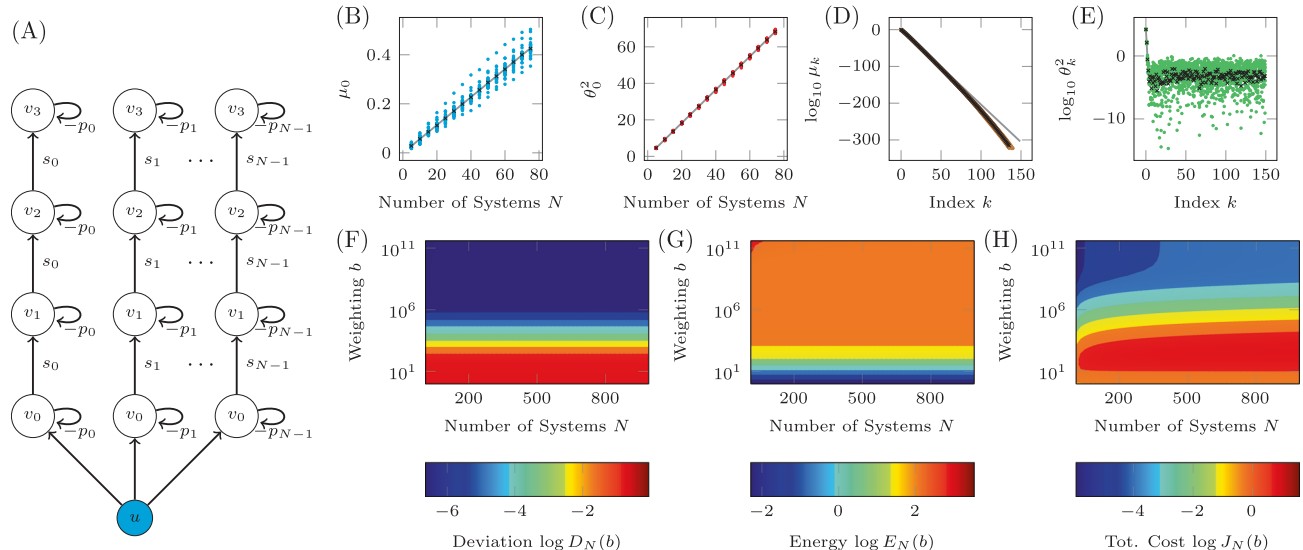

**Fig. 3 An example of the derivations applied to the unidirectional chain graph. A** A diagram of a unidirectional path graph of length $n = 4$ and $N$ possible realizations with loops $-p_k$ and edge weights $s_k$, $k = 0, ..., N-1$. **B** The largest eigenvalues of the COCG when we choose $N$ realizations, where we see the linear growth with $N$. **C** The associated inner products $\theta_0^2 = \xi_0^T \beta$, which is also seen to grow linearly. **D** The eigenvalues for a particular value of $N$ are seen to decay exponentially. Other choices of $N$ lead to nearly the same decay rate $r_1$. **E** The associated eigenvectors multiplied by the control maneuver where we see the exponential decay initially for $k < \bar{k}$ and then saturation for $k > \bar{k}$ where $\bar{k} = 4$ for this choice of $N$. **F** The log average deviation, $D_N(b)/Np$, as a function of $N$ and $b$ computed using the values found for $c_1$, $c_2$, $r_1$, $r_2$, and $\theta_c^2$. **G**, **H** The log control energy and the log total cost as functions of $N$ and $b$, respectively.

Eq. (11) as well as that assumptions 1 and 2 hold. As a first example, we consider the simplest possible network, a unidirectional path graph $\mathcal{G} = (\mathcal{V}, \mathcal{E})$ which consists of $|\mathcal{V}| = n$ nodes, labeled $v_j$, $j = 0, ..., n-1$, and directed edges $(v_j, v_{j+1}) \in \mathcal{E}$, $j = 0, ..., n-2$. There is a uniform loop weight at each node of weight $-p$ and uniform edge weight $s$. The control input matrix $B = e_0$ assigns the single control input to node $v_0$. The loop weight and the edge weight are assumed to be uncertain, but be drawn from distributions, from which we sample $N$ adjacency matrices $A^{(k)}$, $k = 0, ..., N-1$. Each adjacency matrix, $A^{(k)}$, is a bidiagonal matrix with $-p_k$ along the main diagonal and $s_k$ along the first subdiagonal. To describe the matrix $B$ and $C$, we define two sets of nodes; driver nodes $\mathcal{D} \subseteq \mathcal{V}$ and target nodes $\mathcal{T} \subseteq \mathcal{V}$. The set of $|\mathcal{D}| = m$ driver nodes can be represented as the matrix $B$ where each column of $B$ has a single non-zero element corresponding to the index of a driver node. The set of $|\mathcal{T}| = p$ target nodes describes the nodes whose states we are interested in driving to a particular value at the final time, $t = t_f$. The output matrix $C$ consists of $p$ rows where the sole non-zero entry in each row corresponds to the index of a target node[19].

An example of the uncertain unidirectional chain graph is shown in Fig. 3(A) where the single input, labeled $u$ and colored blue, is connected to the $N$ copies of the driver node $v_0$. Each copy of node $v_j$ is connected to the corresponding copy of the node $v_{j-1}$, $j > 0$. The simplicity of this network and choice of only two unknown weights removes many of the other complicating factors, reducing the problem to only 3 variables; the distribution from which the loop weights are drawn, $\mathcal{P}_p$, the distribution from which the edge weights are drawn, $\mathcal{P}_s$, and the choice of target nodes $\mathcal{T} \subseteq \mathcal{V}$. An example of the four expressions in Assumption 1 and 2 are shown in Figs. 3 (B)-(E). For these simulations, $p \in \mathcal{U}(2,4)$ and $s \in \mathcal{U}(0.5, 1.5)$, where $\mathcal{U}(a,b)$ is the uniform distribution between $a$ and $b$. The set of target nodes in this case is only $\mathcal{T} = \{v_1\}$ and $y_f = 1$. The results shown here are qualitatively the same for other choices of distributions and/or set of target nodes, with the only difference being the rates of growth or decay, $c_1$, $c_2$,

$r_1$, $r_2$, and $\theta_c^2$, as laid out in Assumption 1 and Assumption 2. In Fig. 3B, the largest eigenvalue of the COCG, $\mu_0$, is shown to grow linearly with the number of systems $N$ where the blue marks are computed from 10 realizations for each value of $N$, the black marks are the average largest eigenvalue and the gray line is the linear fit computed for the original data. Similarly, in Fig. 3C, $\theta_0^2 = \xi_0^T \beta$ is also shown to grown linearly with $N$. Additionally, the eigenvalues are seen to decay exponentially as stated in Assumption 1, which is shown in Fig. 3D. We also see from Fig. 3E that the values $\theta_k^2$ decay exponentially for $k < \bar{k}$ while they are approximately constant for $k > \bar{k}$. We emphasize that the flooring of $\theta_k^2$ for $k > \bar{k}$ is not a numerical artifact, as all of our calculations are performed by using tunable numerical precision and by verifying accuracy of the results[49–51].

As both Assumptions 1 and 2 hold, we can be sure that the deviation, the control energy, and the total cost remain bounded in the $N \to \infty$ limit. The values used in Assumptions 1 and 2 are found to be approximately $c_1 = 5.70 \times 10^{-3}$, $c_2 = 0.911$, $r_1 = 10^{-2.04}$, $r_2 = 10^{-3.14}$, and $\theta_c^2 = 10^{-6.32}$ (as shown in Fig. 3B–E). The deviation, control energy, and total cost as a function of both $N$ and $b$ (as it appears in Eq. (10)) are shown in Fig. 3F–H, respectively. We see that as $N$ grows there is little change in the deviation or the control energy, while as $b$ grows, the deviation decreases and the control energy increases. In both cases, there is a range of $b$ where the deviation and control energy change rapidly, while for very large $b$ the rate of change decreases rapidly. The total cost grows monotonically as a function of $N$, while it appears that as $b$ grows, there is at least one maximum. These plots are qualitatively similar to those made regardless of the distributions for the regulation $p_k$ and the edge weights $s_k$ or the set of target nodes, where alternative choices only lead to different values of $c_1$, $c_2$, $r_1$, $r_2$, and $\theta_c^2$.

Recently, it was shown that the graph distance between driver nodes and target nodes is an extremely important property when determining the control energy for single system realizations[52,53]. A study on the effect of uncertainty on results previously derived

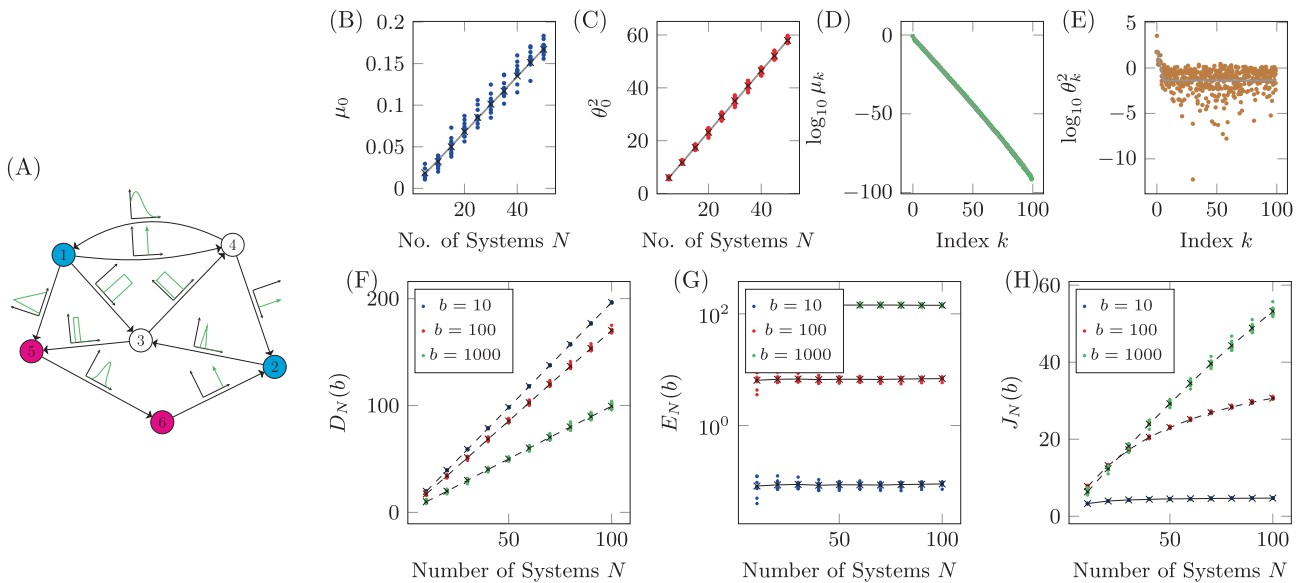

**Fig. 4 A network with uncertain edge weights. A** The diagram of the network with six nodes and 10 edges. Each edge weight is drawn from the distribution shown in the associated plot, with numerical values for the distributions shown in Table 1. Additionally, there is a negative self-loop assigned to each node drawn from the uniform distribution $\mathcal{U}(2, 4)$. The largest eigenvalues $\mu_0$ and associated values $\theta_0^2$ are shown as marks in **B** and **C**, respectively, where the average for each $N$ is shown as a black cross and the gray lines are linear fits. For $N = 50$, the full spectrum of $\mu_k$ is plotted in **D** and the corresponding values $\theta_k^2$ are plotted in **E**. The gray lines represent the curves found after fitting the expressions in Assumptions 1 and 2. The cost terms (deviation, control energy, and total cost) for $b = 10$, $b = 100$, and $b = 1000$ are shown in panels **F**–**H**, respectively. The black marks are the averages over the 10 realizations for each value of $N$ and the dashed lines are for the viewer.

**Table 1 The distributions from which the edge weights are drawn for the graph in Fig. 4A.**

| Edge | Distribution | Edge | Distribution |
|------|-------------|------|-------------|
| (1,4) | $\delta(0.5)$ | (1,3) | $\mathcal{U}(0.5, 1)$ |
| (1,5) | $\mathcal{T}(0.1, 1, 0.3)$ | (2,3) | $\mathcal{N}(0, 1, 0.5, 0.1)$ |
| (3,4) | $\delta(1)$ | (3,5) | $\mathcal{U}(0.1, 0.3)$ |
| (4,1) | $\mathcal{T}(0.2, 0.4, 0.3)$ | (4,2) | $\mathcal{N}(0, 1, 0.2, 0.2)$ |
| (5,6) | $\delta(0.75)$ | (6,2) | $\mathcal{U}(0.2, 0.4)$ |

The edge weights may be constant, $\delta$, drawn from a uniform distribution, $\mathcal{U}$, drawn from a triangular distribution, $\mathcal{T}$, or drawn from a truncated normal distribution, $\mathcal{N}$.

which state that control energy grows exponentially with distance between a single driver node and a single target node is presented in Supplementary Note 2. The next model we consider is a linear system which can be described by a network where the edge weights are drawn from distributions assigned to each edge. An example of this kind of network is shown in Fig. 4A where the distributions each edge weight is drawn from are shown qualitatively along the edges with further details collected in Table 1. We choose delta distributions for three edges which represents the case where an edge weight is known exactly, uniform distributions for three edges, triangular distributions for two edges, defined as $\mathcal{T}(a, b, c)$ where $a < c < b$ and truncated normal distributions for the remaining two edges. There is a negative self-loop at each node drawn from a uniform distribution $\mathcal{U}(2, 4)$.

For this network, we choose nodes 1 and 2 to be driver nodes and nodes 5 and 6 to be target nodes so that $B = [I_2 O_{2 \times 4}]^T$ and $C = [O_{2 \times 4} 7D1I_2]$. The final vector value is chosen to be $y_f = [17D11]^T$ and $t_f$ is chosen to be large enough such that $e^{A^{(k)} t_f} x_0$ is sufficiently close to zero to be ignored. The largest eigenvalue, $\mu_0$, and associated values $\theta_0^2$, as a function of $N$, are shown in Fig. 4B, C where we see the linear increase required by Assumptions 1 and 2. For $N = 50$, all of the eigenvalues, $\mu_k$, and associated values $\theta_k^2$,

for 25 realizations, are shown in Figs. 4D, E, respectively. Again, it is apparent that the behavior agrees with the requirements laid out in Assumptions 1 and 2. As both assumptions hold, we can be sure that $D_N(b)/Np$, $E_N(b)$, and $J_N(b)$ all approach constant values in the $N \to \infty$ limit. The particular values approached in this limit depend on the choice of $b$. The deviation is shown in Fig. 4F and the control energy is shown in Fig. 4G. We see that, since $\frac{\partial D_N(b)/Np}{\partial b} < 0$, as $b$ grows, the slope of the deviation decreases. Similarly, since $\frac{\partial E_N(b)}{\partial b} > 0$, as $b$ grows, so does the control energy. Finally, the total cost is shown in Fig. 4H, where the different growth rates are due to the coefficient $\frac{Np}{Np+b}$ that appears in the approximate expressions derived in Supplementary Note 1.4.

Again, alternative choices of distributions for each edge weight and loop weight, sets of target nodes, and sets of drivers nodes, lead to qualitatively similar plots as shown in Fig. 4 except that the particular rates of increase, or constant values, will change. A common control goal is driving a nonlinear system near one of its fixed points using its linearization. Even for the case the system is not near a fixed point, the linearization can be used in a piecewise manner as discussed in Kilckstein et al.[20]. Generically, a controlled nonlinear system is written as,

$$\dot{x}(t) = f(x(t), u(t); \phi) \qquad (12)$$

where we assume there are $n$ states, $x_j(t)$, $j = 1, \dots, n$, and $m$ control inputs, $u_j(t)$, $j = 1, \dots, m$ and some parameters collected in $\phi$. Near a fixed point, $(\bar{x}, \bar{u})$, such that $f(\bar{x}, \bar{u}; \phi) = 0$, then the behavior of the system is approximately,

$$\delta \dot{x}(t) = A \delta x(t) + B \delta u(t) \qquad (13)$$

where $\delta x(t) = x(t) - \bar{x}$ and $\delta u(t) = u(t) - \bar{u}$ are the states and inputs relative to the fixed point and $A = \frac{\partial f}{\partial x}\big|_{x=\bar{x}}$ and $B = \frac{\partial f}{\partial u}\big|_{u=\bar{u}}$ are the Jacobians of $f$ relative to the states $x$ and the inputs $u$, respectively, evaluated at the fixed point. The resulting linearized system can be represented as a network, where directed edges exist between states $x_j$ and $x_k$ if $\frac{\partial f_j}{\partial x_k} \neq 0$. Note that the fixed point

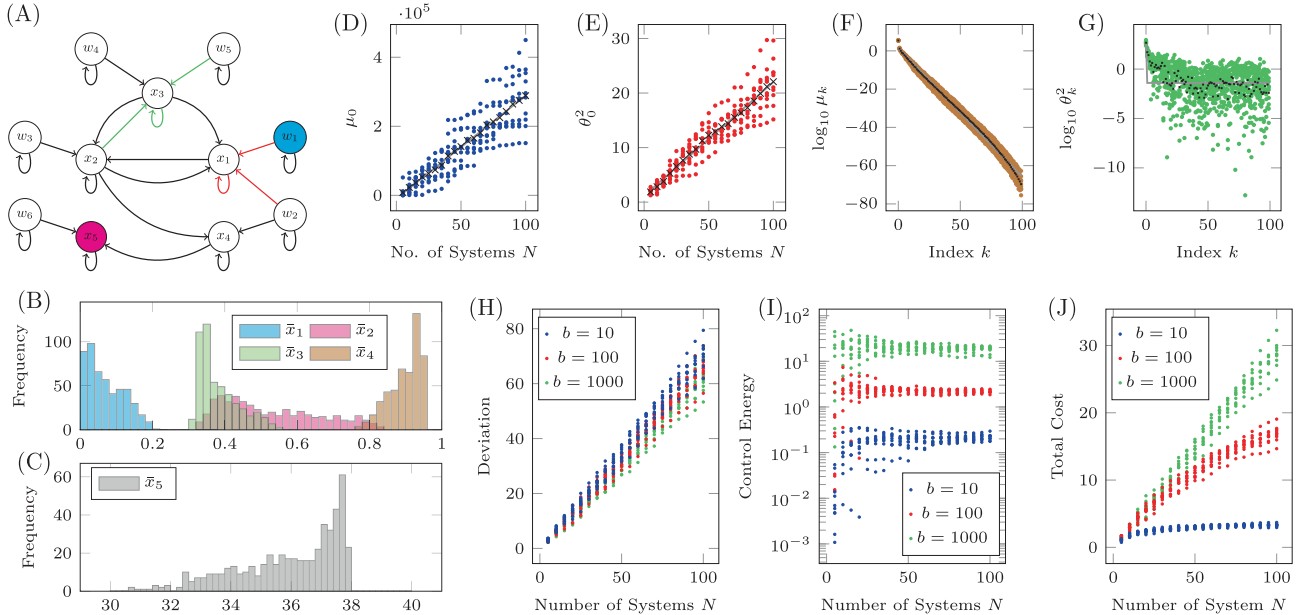

**Fig. 5 The Jacobian of a system with uncertain parameters. A** The Jacobian of the simplified model of autophagy represented as a network. Red edges have weights in which $C_{NU}$ appears explicitly and green edges have weights in which $C_{EN}$ appears explicitly, while black edges have weights that may or may not implicitly depend on $C_{EN}$ and $C_{NU}$. For 500 choices of $C_{NU}$ and $C_{EN}$, the stable fixed point is computed and collected in the bar plots in **B** (for $\bar{x}_k$, $k = 1, 2, 3, 4$) and **C** (for $\bar{x}_5$). Note that even though $C_{NU}$ and $C_{EN}$ are drawn from uniform distributions, the values of the fixed point are not unfiformly distributed. The largest eigenvalue $\mu_0$ and associated value $\theta_0^2$ are shown in **D** and **E** for 10 realizations of $N$ random choices of $C_{NU}$ and $C_{EN}$. For 10 realizations of $N = 100$, the complete eigendecomposition, $\mu_k$ and $\theta_k^2$, are shown in **F** and **G** where Assumptions 1 and 2 are seen to hold. The resulting deviation, control energy, and total cost are shown in **H**–**J**, respectively.

$(\bar{x}, \bar{u})$ depends upon the particular set of parameters $\phi$, and so the matrices $A$ and $B$ also depend on the choice of $\phi$. If the system of interest represents something for which taking measurements is difficult, often many of the parameters are only know approximately and so any controller derived using one particular set of control inputs is not guaranteed to be satisfactory for a different set.

As an example of this type of system, we apply our methodology to a recently published model of autophagy in cells[22]. The model contains five internal states which represent the properties of the cell itself, labeled $x_1$ through $x_5$, and six auxiliary states that represent the current concentration of drugs which may be introduced to the cell, labeled $w_1$ through $w_6$. This model consists of dozens of parameters but here we consider two in particular, $C_{EN}$ and $C_{NU}$, which are coefficients that represent the amount of energy and nutrients available in a cell. As these parameters are cell dependent, their particular values may vary across multiple cells. This model was shown to have a stable fixed point for a range of values of $C_{EN}$ and $C_{NU}$. We assume that all that is known about $C_{EN}$ and $C_{NU}$ is that they both lie between 0.1 and 0.6. The model is linearized about the stable fixed point and the resulting network is shown in Fig. 5A. In this system, we are interested in adjusting the amount of drug of type 1 (making $w_1$ the sole driver node) to regulate the level of autophagy (making $x_5$ the sole target node) which are color coded accordingly.

The fixed point of the system, about which the linearization is performed, is computed for 500 random choices of $C_{EN}$ and $C_{NU}$ selected uniformly from $\mathcal{U}(0.1, 0.6)$ and the resulting values are binned in Fig. 5B, C. Note that despite the parameters being drawn from uniform distributions, the fixed points are clearly not uniformly distributed in state space. Additionally, we see in Figs. 5D, E that $\mu_0$ and $\theta_0^2$ grow approximately linearly with $N$ while in Fig. 5F, G the eigenvalues $\mu_k$ decay exponentially and $\theta_k^2$ initially decay before saturating, thus Assumptions 1 and 2 hold.

Note that $\mu_0 \sim 10^5$ for the range of $N$ shown, much larger than the previous examples, but this does not affect the validity of our derivations. As the assumptions hold, we can be sure of the following three states; (i) that the deviation grows linearly with $N$ regardless of the choice of $b$ which is shown in Fig. 5H, (ii) the control energy approaches a constant value, seen in Fig. 5I, and (iii) the total cost approaches a constant, seen in Fig. 5J, for $b = 10$, $b = 100$, and $b = 1000$.

Qualitatively similar results can be seen for alternative choices of therapy, that is, rather than choosing only drug 1, one could instead choose any combination of the six drugs. Also, if more information is known about the probability of $C_{NU}$ and $C_{EN}$, then alternative distributions can be chosen from which these parameters are drawn. Next we examine the relationship between the number of target nodes and the cost. We have seen that controlling network ensembles requires more control energy than controlling a single network realization. Here we investigate the relationship between the number of target nodes and the energy required for controlling the ensemble. We see that in average the control energy decreases exponentially, as the number of target nodes is reduced, which indicates feasibility of our approach, as long as the number of target nodes remains small. To demonstrate this relationship, for each realization of $N$ uncertain systems, $b$ is chosen such that $D_N(b)/(Np)$ is a constant value regardless of the set of target nodes. To find $b$, bisection is used as $D_N(b)$ monotonically decreases with $b$. The values of $b$ are averaged over target sets of the same cardinality in Fig. 6A and are seen to grow exponentially as the set of target nodes only grows linearly. The desired deviation is seen to be achieved in Fig. 6B where the error bars are smaller than the size of the marks as the bisection tolerance was set to $10^{-16}$. The resulting control energies are collected and their geometric mean is taken over sets of target nodes of the same cardinality in Fig. 6C. We see that as the cardinality of the target node set, $|\mathcal{T}|$, decreases linearly, the

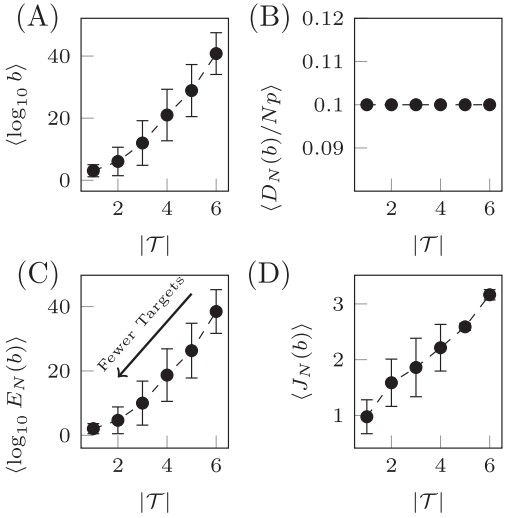

**Fig. 6 Target control of network ensembles.** The costs (weighting term $b$, deviation $D_N(b)$, control energy $E_N(b)$, and total cost $J_N(b)$) averaged over sets of target nodes of the same cardinality for the small network shown in Fig. 4A. The weighting term $b$ is chosen such that $D_N(b)/(Np) = 0.1$ and the result is shown in panel **A**. The deviation is shown in panel **B** where the desired value is seen to be achieved. In **C**, the control energy is shown where it is clear as the number of target nodes decrease, the control energy decreases exponentially. The total cost in **D** is seen to grow approximately linearly. Error bars represent standard deviation.

geometric mean of the control energy decays exponentially, leading to the conclusion that small reductions in the set of target nodes can lead to immense reductions in effort. Finally, the total cost is shown in Fig. 6D which is seen to decrease linearly as the number of target nodes is reduced. This can be explained as a result of our choice to hold $D_N(b)/(Np)$ constant which leads to $b \approx E_N(b)$ so $J \sim Np$. We would like to emphasize that these results for network ensembles differ from our previous work[19], in which we had reported a similar scaling relationship for single network realizations, but for the case that the control goal had a constrained final position, while here we are allowing some deviation from the desired final state.

## Discussion

The lack of precise information about the mathematics behind many biological systems motivated us to study optimal control of uncertain systems represented by network ensembles, where each edge weight is drawn from a given distribution rather than being exactly known. A practical application of our analysis is an experimental situation in which some of the system parameters are known to lie in a bounded range, but their exact value is unknown. In the presence of such uncertainty, we are able to analytically solve an associated optimal control problem. We have characterized the solution to this problem in the limit of infinitely many system realizations corresponding to the case the realizations are drawn from a continuous distribution. We have then showed how to properly formulate the objective function to ensure feasibility of the problem as the number of realizations grows.

We first demonstrated the feasibility of controlling uncertain linear systems, for the case that the state matrix $A$ may be one of $N$ possible choices drawn from some possibly continuous distribution such that the deviation, or variance, of the final state around some desired final state is maintained below a desirable threshold. We then extended this analysis to nonlinear systems

with uncertain parameters. An example of such a system studied in the paper is the cell regulatory network of autophagy. We assumed that the amounts of energy and nutrient available to the cells were uncertain, which yielded different fixed points for the dynamics, about which the equations were linearized. We then characterized the optimal cost of controlling more and more realizations of this network (each one corresponding to different levels of available nutrient and energy). As long as the two assumptions about the COCG hold, which we have found to be the case for all systems analyzed, from simple networks to linearizations of complicated nonlinear dynamical networks, we have analytically shown that the average deviation and the control energy remain finite in the $N \to \infty$ limit. This implies the feasibility of deriving a control input, not for a particular system, but rather for a system described only in terms of distributions, possibly determined experimentally.

Our main result is that as long as the weighting parameter $\alpha(N)$ is chosen properly, the cost of the optimal control solution remains finite. The price to pay for controlling uncertain systems is a higher cost of the optimal control solution. However, this cost can be consistently (exponentially) reduced by limiting the number of target nodes, i.e., the nodes chosen as targets of the control action.

## Methods

**Multiple precision**. To check assumptions 1 and 2, we required an ability to compute eigenvalues with additional accuracy not possible using double precision as they will typically be extremely small. To do this, we implement a few numerical methods with the multiple precision data type provided in the MPFR library[50] which is built on top of Gnu GMP[49]. Additionally, for multiple precision complex variables, we use the extension to MPFR called MPC[51]. The code which we use to perform the simulations contained in the text is available at the following Github repository.

**Sylvester equations**. To find each block of the COCG as defined in Eq. (5), we solve the Sylvester equation,

$$A^{(j)}W_{j,k} + W_{j,k}A^{(k)^T} = -BB^T, \quad j,k = 0, \dots, N-1 \tag{14}$$

where we assume $A^{(j)}$ is negative definite. Let $V^{(j)}$ and $D^{(j)}$ be the complex matrix of eigenvectors and eigenvalues, respectively, of the $j$th matrix $A^{(j)}$ so that

$$A^{(j)}V^{(j)} = V^{(j)}D^{(j)} \tag{15}$$

Then, applying the eigenvector transformation in Eq. (15) to the Sylvester equation in Eq. (14) yields the solution,

$$W_{j,k} = V^{(j)}\left(Y_{j,k} \circ \left(V^{(j)^{-1}}BB^T V^{(k)^{-T}}\right)\right)V^{(k)^T} \tag{16}$$

where the matrix $Y_{j,k}$ has elements equal to the inverse $\frac{1}{d_a^{(j)}+d_b^{(k)}}$ where $d_a^{(j)}$ and $d_b^{(k)}$ are the $a$th and $b$th eigenvalue of $A^{(j)}$ and $A^{(k)}$, respectively. The eigenvalues and eigenvectors are determined using a real Schur decomposition of each $A^{(j)}$ to reduce it to upper Hessenberg form with a unitary transformation. This is accomplished using the QR iteration described in Chapter 7 in Golub and Van Loan[54] where the eigenvectors are recovered from the corresponding Schur vectors. Once the eigenvectors are known, we must solve the complex non-Hermitian systems of equations $V^{(j)}B^{(j)} = B$ which appear in Eq. (16). The LU decomposition of each eigenvector matrix is computed as described in Chapter 3 of Golub and Van Loan[54] and stored as each matrix $B^{(j)}$ will appear in $N$ blocks $W_{j,k}$, $k = 0, \dots, N-1$. The entire COCG is compiled by pre- and post-multiplying each block $W_{j,k}$ by $C$ and $C^T$, respectively.

**Symmetric matrix eigen-problems**. Once the complete COCG is available, we are interested in computing the total eigendecomposition. As the COCG is real and symmetric, we use a symmetric tridiagonal decomposition using Householder matrices. Once the symmetric tridiagonal matrix is available, we can use QR steps again to determine the eigenvalues, as well as we can recover the eigenvectors from the Householder matrices as described in Chapter 8 in Golub and Van Loan[54].

To compute the costs more efficiently than using the eigendecomposition, we use the quadratic form in Eq. (8). This requires solving the linear system in Eq. (7) which is a symmetric positive definite system of equations. The Cholesky decomposition of $\bar{U}(\alpha)$ is computed in order to find the optimal distance away from the desired distance $\gamma$. The procedure we implement is described in Chapter 4 of Golub and Van Loan[54].

## Data availability
Data for each of the figures is available upon request.

## Code availability
Code used to generate data is available from the online repository https://github.com/iklick/controlling_network_ensembles

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

## Acknowledgements
This work has been supported by the National Science Foundation through grants No. 1727948 and No. CRISP- 1541148. The authors thank Franco Garofalo, Francesco Lo Iudice, and Anna Di Meglio for insightful discussions during the development of this problem.

## Author contributions
F.S. proposed the problem; I.K. developed the theoretical results and performed the numerical studies; I.K. and F.S. wrote the paper.

## Competing interests
The authors declare no competing interests.
