## [Peer Review File · Nature Communications]

Reviewer #1 (Remarks to the Author):

The present work is highly innovative as it characterizes control of dynamics defined on network ensembles, i.e. networks in which there is only statistical and partial information about the network.

The traditional theory of control applies to given networks, structural controllability has extended this approach by considering weighted adjacency matrix with unknown weights.

Now the authors of this work also relax the constraint on the given and fixed topology of the network.

The work is solid, and good, however it is very technical so I would suggest to give more background and more narrative to be more accessible to Nature Communication typical readers.

Also I have two points to make:

1) the neural network of e.coli is actually quite well established although there might be different genomic variation. So people use it as brain network as a consensus model. The metabolic network is quite well established and fixed. The protein-protein interaction network is widely unknown because it is undersampled. It would be good if these cases are treated differently in the introduction because the source of statistical error can be very different.

2) If I understand correctly the network model considered here are canonical not microcanonical, i.e. they do not have fixed hard constraint like the degree sequence by soft constraint... Maybe I am mistaken by weighted ensemble do not exist in the microcanonical version as far as I know. Given the novelty of the work I think that the work can be published in Nature Communication provided the authors address my concerns and improve the presentation to make the paper more accessible to the general reader of Nature Communications.

Reviewer #2 (Remarks to the Author):

This paper studies the problem of the optimal control of an uncertain system using Monte Carlo sampling of its unknown parameters and designing the control input such that the final state deviation of some average system is optimized. As stated by the authors, their main result is that the optimal cost remains finite as the number of Monte Carlo samples goes to infinity, thus providing better and better approximations of the uncertainty distributions.

I like that the paper is clearly written and the contributions are stated clearly. The flow of the presentation of the preliminaries and results is also very good. The topic is also very timely, with applications all over science and engineering. There are also two points that are not major contributions but I liked about the paper. One is the formulation of the problem in the form of the optimization problem (4). It is quite straightforward but still not something that I had seen elsewhere. Second is the exponential decay rates they have observed as Assumptions 1 and 2. They are not proved or shown in any systematic form, only in 3 examples, but seeing them formulated in one place is nice.

What I am most concerned about is the significance of the contributions. The main conclusion of the paper that the optimal cost remains finite seems trivial to me, unless I am missing something. The reason is that, let's assume the control input u is set to 0 for all t . As long as the control horizon is finite, which is the case in this paper, the deviation of the output $Cx_j(t)$ from the target output y_f is necessarily finite (say U_j), no matter how large, even if the system is unstable. It is of course true that this finite value U_j is different for different system realizations j , but the authors' choice of $\alpha(N)$ is such that $(1-\alpha)^*D$ is essentially a Monte Carlo (sample) expected value of the output deviation, which is expected to be finite especially when the parameters have finite support. Even if the support of parameter distributions was unbounded, still I'd expect the expected value of output deviation to be finite for finite time horizon. And all of these were assuming $u = 0$. When you can optimize over u , the bound will definitely be lower. So in any case, I do not see why this main result of the paper is non-trivial. Also, I am not sure how the result is useful, because the upper bounds on the optimal cost can be arbitrarily high. They are only finite, which I don't think is enough for any application.

I would recommend giving the authors a chance to revise, but if granted, I only advise them to use it if they have a strong response in proof of the significance of their contributions. If they want

to resubmit, please also address the following. I also haven't checked all the derivations carefully at this point, and will do so later if a revision is to be made. So far, the derivations that I have checked have been correct except for minor modifications mentioned below.

1) There is quite some literature on robust optimal control. I do not see any papers from that literature cited, even though the problem of robust optimal control is almost exactly the same as this paper: finding the optimal control of a system with (parametric) uncertainty. There might be some differences in approach, but the distinction and superiority with respect to that literature should be comprehensively discussed.

2) The abstract reads "Our work sheds fundamental insight into the relationship between optimality and uncertainty." Please either remove that sentence or clearly explain what fundamental light the work is shedding into this relationship. This is of course related to my main concern above.

3) The introduction reads "While the optimal control can be derived for any particular set of parameters, the resulting control is only optimal for that set." How is this not exactly the case for this paper? If N is large you can intuitively say that the optimal control of the finite sample is also probably good for the entire distribution, but you haven't shown that, and the same would hold for any other work. Note that in contrast, the literature of robust optimal control I mentioned often consider infinite parameter sets all at once (at least as far as I know), not using finite sampling as done here.

4) The construction of the composite system has a minor problem and that is the fact that the matrix \tilde{C} is not a horizontal concatenation of N copies of C , but a block-diagonal concatenation such as in \tilde{A} . The vector $y(t)$ should also be defined the same way as $x(t)$.

5) The two matrices \bar{U} and \bar{W} are NOT similar. Two matrices A and B are similar if they can be related to one another via a similarity transform $A = PBP^{-1}$. Similar matrices always have the same eigenvalues but different eigenvectors.

6) The ordering of θ_k (with k) in Assumption 2 cannot be the same as the ordering of μ_k in Assumption 1. This is because the ordering of θ_k can be changed arbitrarily by changing y_f . Please clarify this in the paper.

7) The matrix B is defined in two inconsistent ways in Example 1. One is for what shown in the main text and one is more general for the supplementary.

8) What is the point of assuming that the parameter supports be finite? I don't see any use of it in the paper.

9) In the supplementary, the sentence following (S8) is incorrect. There is no optimization problem remaining. There is only a set of equations to solve to find $x(t_f)$. That what the authors do as well, solve for $x(t_f)$ from the constraints, no free variable is left after applying all the conditions of maximum principle.

**Controlling network ensembles**

**Response to Reviewers**

Isaac Klickstein^{1, a)} and Francesco Sorrentino^{1, b)}

*Department of Mechanical Engineering, University of New Mexico, Albuquerque,*

*NM 87131*

^{a)}Electronic mail: iklick@unm.edu

^{b)}Electronic mail: fsorrent@unm.edu

**I. REVIEWER 1**

**The present work is highly innovative as it characterizes control of dynamics defined on**
**network ensembles, i.e., networks in which there is only statistical and partial information**
**about the network. The traditional theory of control applied to given networks, structural**
**controllability, has extended this approach by considering a weighted adjacency matrix with**
**unknown weights. Now the authors of this work also relax the constraint on the given and**
**fixed topology of the network. The work is solid and good however it is very technical so I**
**would suggest to give more background and more narrative to be more accessible to Nature**
**Communication's typical readers.**

We thank the reviewer for finding our work highly innovative and solid. We have worked on
the text to make it more accessible to the broad readership of Nature Communications. In partic-
ular, we have rewritten much of the first part of the results section to better emphasize our main
results and relegated some of the details to the supplementary information. These changes are
highlighted in red. We hope the reviewer will find that the presentation of the paper has now
improved.

**Also I have two points to make;**

- **1. The neural network of *e.coli* is actually quite well established although there might be**
**different genomic variation so people use it as a brain network as a consensus model.**
**The metabolic network is quire well established and fixed. The protein-protein inter-**
**action network is widely unknown because it is undersampled. It would be good if**
**these cases are treated differently in the introduction because the source of statistical**
**error can be very different.**

We thank the reviewer for pointing to us how different sources of statistical error affect dif-
ferent *E. coli* networks published in the literature. In the revised paper we have mentioned
the differences between metabolic and protein-protein networks and added appropriate ref-
erences. For the metabolic network, it is known that the steady state rates of transition
between metabolites can be affected by the environment¹. If the environment is known
within some range, then a set of transition rates can be found to create the different realiza-

tions. This point is emphasized in the revised manuscript.

**2. If I understand correctly, the network model considered here are canonical, not mi-**
**crocanonical, i.e., they do not fix hard constraints like the degree sequence by soft**
**constraint... Maybe I am mistaken as weighted ensembles do not exist in the micro-**
**canonical version as far as I know.**

Yes, the reviewer is correct. In our examples, we do not impose hard constraints on the
networks, such as that they must satisfy a specific degree sequence, though in principle we
could apply our analysis to that case. In two of our examples, the existence of the connec-
tions is assigned but the weights associated to the connections are not, which indicates the
ensemble constraints are satisfied on average. Thus based on the reviewer's input, we think
it is more appropriate to call the ensembles we consider in the paper canonical, with the
more specific case of microcanonical ensembles remaining a possibility.

**Given the novelty of the work I think the work can be published in Nature Communication**
**provided the authors address my concerns and improve the presentation to make the paper**
**more accessible to the general reader of Nature Communications.**

We thank the reviewer for their positive feedback. We have followed their recommendation
and worked on the presentation to make it more accessible to the broad readership of Nature
Communications. All the changes aimed at increasing readability have been highlighted in red.

**II. REVIEWER 2**

This paper studies the problem of the optimal control of an uncertain system using Monte
Carlo sampling of its unknown parameters and designing the control input such that the fi-
nal state deviation of some average system is optimized. As stated by the authors, their main
result is that the optimal cost remains finite as the number of Monte Carlo samples goes to
infinity, thus providing better and better approximations of the uncertainty distributions.

I like that the paper is clearly written and the contributions are stated clearly. The flow
of the presentation of the preliminaries and results is also very good. The topic is also very
timely, with applications all over science and engineering. There are also two points that are
not major contributions but I liked about the paper. One is the formulation of the problem
in the form of the optimization problem (4). It is quite straightforward but still not some-
thing that I had seen elsewhere. Second is the exponential decay rates they have observed
as Assumptions 1 and 2. They are not proved or shown in any systematic form, only in 3
examples, but seeing them formulated in one place is nice.

We thank the reviewer for their encouraging comments and for mentioning the novelty and time-
liness of our approach.

What I am most concerned about is the significance of the contributions. The main con-
clusion of the paper that the optimal cost remains finite seems trivial to me, unless I am
missing something. The reason is that, let's assume the control input $u(t)$ is set to 0 for all t .
As long as the control horizon is finite, which is the case in this paper, the deviation of the
output $Cx_j(t)$ from the target output y_f is necessarily finite (say U_j), no matter how large,
even if the system is unstable. It is of course true that this finite value U_j is different for
different system realizations j , but the authors' choice of $\alpha(N)$ is such that $(1 - \alpha)D$ is essen-
tially a Monte Carlo (sample) expected value of the output deviation, which is expected to be
finite especially when the parameters have finite support. Even if the support of parameter
distributions was unbounded, still I'd expect the expected value of output deviation to be
finite for finite time horizon. And all of these were assuming $u = 0$. When you can optimize
over u , the bound will definitely be lower. So in any case, I do not see why this main result

**of the paper is non-trivial. Also, I am not sure how the result is useful, because the upper**
 **bounds on the optimal cost can be arbitrarily high. They are only finite, which I don't think**
 **is enough for any application.**

The reviewer is bringing up an important question, related to the underlying significance of
 our contribution. We are grateful for having a chance to clarify this point. The case the author
 is presenting ($u = 0$) is not actually encompassed in our formulation Eq. (4). In fact, this case
 is a solution when setting $\alpha = 1$ (we only allow α to vary in the open interval $(0, 1)$). In the
 revised version of the paper we have added a sentence that explicitly mentions that the $\alpha = 1$
 case is excluded.) For the special case $\alpha = 1$, we have that the objective function coincides with
 only the control energy $J = \frac{1}{2}E$, for which the solution of the optimal control problem simply
 yields $u = 0$ and the objective function achieves a minimum at $J = 0$. However, for the case we
 consider $0 < \alpha < 1$, things can be quite different, because the objective function depends on both
 the deviation and the control energy.

The reviewer is correct that, for finite time, $D_N(\alpha)/Np$ remains finite even in the thermody-
 namic limit ($N \rightarrow \infty$). This is both clear from the logical arguments made by the reviewer, and is
 proven in Eq. (S37) in the revised SI where we show the average deviation is bounded by a value
 independent of the weighting parameter α and converges to a finite value in the thermodynamic
 limit. What we found to not be trivial is that for proper choice of $\alpha(N)$, the control energy remains
 bounded in the thermodynamic limit since this does not hold true for other choices of $\alpha(N)$. In
 the revised SI, in Eq. (S29) we derive the approximate control energy, $\bar{E}_N(\alpha)$, and bound it on
 both sides with,

$$112 \quad \bar{E}_{N,LP}(\alpha) < \bar{E}_N(\alpha) < \bar{E}_{N,UB}(\alpha) \quad (1)$$

where the lower bound appears in Eq. (S34) and grows as $\frac{1}{Np} \left(\frac{1}{r_1}\right)^{\log\left(\frac{(1-\alpha)Np}{\alpha}\right)}$ (noting $0 < r_1 < 1$)
 while the upper bound appears in Eq. (S30) grows as $\left(\frac{(1-\alpha)Np}{\alpha}\right)^2$. Note that both the lower and
 upper bounds' behavior depends on the quantity,

$$116 \quad \frac{(1-\alpha)Np}{\alpha} \quad (2)$$

and its behavior as $N \rightarrow \infty$. For the upper and lower bounds to remain finite in the thermodynamic
 limit, it is necessary that,

$$119 \quad \lim_{N \rightarrow \infty} \frac{(1-\alpha(N))Np}{\alpha(N)} = b < \infty \quad (3)$$

Note that this limit does not exist if α is constant or if $\alpha(N)$ does not approach 1 in the thermody-
 namic limit. We choose $\alpha(N) = \frac{Np}{Np+b}$ so that the quantity $\frac{(1-\alpha)Np}{\alpha} = b$, completely independent of
 N , making our results concerning the tuning of the average deviation simpler. Alternative choices,
 such as $\alpha(N) = \frac{Np-b}{Np}$, leads to $\frac{(1-\alpha)Np}{\alpha} = \frac{bNp}{Np-b}$, leads to unnecessarily complicated results and
 additional requirements on b .

To the second point that the bound can be arbitrarily large, in section S1.5 of the revised SI, we
 demonstrate that the average deviation is bounded (as stated by the reviewer) between,

$$127 \quad 0 < \frac{\bar{D}_N(b)}{Np} < \frac{\bar{D}_{N,UB}(b)}{Np} \quad (4)$$

While this upper bound can not be tuned by adjusting b , we show in the revised SI in Eqs. (S46)
 and (S47) that the approximate average deviation is monotonically decreasing in b and bounded
 below by 0 in the limit of $b \rightarrow \infty$. Thus, there exists a value of b such that $\frac{\bar{D}_N(b)}{Np} \leq \varepsilon$ for any
 positive value ε . The fact we can tune b to achieve any desired upper bound with the trade-off that
 the upper bound of the control energy grows with b^2 is now emphasized in the manuscript.

**I would recommend giving the authors a chance to revise, but if granted, I only advise**
 **them to use it if they have a strong response in proof of the significance of their contribu-**
 **tions. If they want to resubmit, please also address the following. I also haven't checked**
 **all the derivations carefully at this point, and will do so later if a revision is to be made. So**
 **far, the derivations that I have checked have been correct except for minor modifications**
 **mentioned below.**

We hope the reviewer is convinced by the changes we have made to refocus the paper towards the
 fact that the control energy in the thermodynamic limit, $\lim_{N \rightarrow \infty} E_N(\alpha) < C_E$, some constant, which
 can be calculated, while maintaining an upper bound on the deviation which can be designed,

$$144 \quad \lim_{N \rightarrow \infty} D_N(\alpha)/N < C_D.$$

- **1. There is quite some literature on robust optimal control. I do not see any papers from**
 **that literature cited, even though the problem of robust optimal control is almost ex-**
 **actly the same as this paper: finding the optimal control of a system with (parametric)**
 **uncertainty. There might be some differences in approach, but the distinction and**

**superiority with respect to that literature should be comprehensively discussed.**

The problem of integrating optimal control and robustness has been dealt with in a num-
ber of classic works, such as for example²⁻⁸. Typically though, robust control develops
*feedback control* which, for biological systems, may not be feasible with limited ability
for continuous measurement. We have briefly reviewed the field of robust optimal control
and included a discussion of previous work in this area in the introduction. We have also
emphasized the differences with our proposed approach and the reason why we believe it is
convenient when dealing with uncertain biological systems.

- **2. The abstract reads "Our work sheds fundamental insight into the relationship between**
**optimality and uncertainty." Please either remove that sentence or clearly explain what**
**fundamental light the work is shedding into this relationship. This is of course related**
**to my main concern above.**

That sentence referred to the fact that we compute the optimal solution as a function of
an increasing number of system realizations, i.e., increasing uncertainty. However, based
on the reviewer's comment, we have removed this statement from the abstract.

- **3. The introduction reads "While the optimal control can be derived for any particular**
**set of parameters, the resulting control is only optimal for that set." How is this not**
**exactly the case for this paper? If N is large you can intuitively say that the optimal**
**control of the finite sample is also probably good for the entire distribution, but you**
**haven't shown that, and the same would hold for any other work. Note that in contrast,**
**the literature of robust optimal control I mentioned often consider infinite parameter**
**sets all at once (at least as far as I know), not using finite sampling as done here.**

The sentence in the introduction was not referring to the particular optimal control problem
considered in this paper, but to previous work in which several optimal control solutions
were computed for slightly different versions of the same "network" to assess robustness
of the computed optimal control solution. In this previous work, the optimal control inputs
$u_1(t)$ and $u_2(t)$, ... were allowed to vary from system realization to system realization. Then
they were compared to assess the extent to which the optimal control solution was sensitive

to parameter variations.

4. **The construction of the composite system has a minor problem and that is the fact that the matrix \tilde{C} is not a horizontal concatenation of N copies of C , but a block-diagonal concatenation such as in \tilde{A} . The vector $y(t)$ should also be defined the same way as $x(t)$.**

We thank the reviewer for pointing this out. The definition of the matrix \tilde{C} has been corrected.

5. **The two matrices \bar{U} and \bar{W} are *not* similar. Two matrices A and B are similar if they can be related to one another via a similarity transform $A = PBP^{-1}$. Similar matrices always have the same eigenvalues but different eigenvectors.**

The reviewer is correct and we apologize for the oversight. This has been fixed in the revised manuscript.

6. **The ordering of θ_k (with k) in Assumption 2 cannot be the same as the ordering of μ_k in Assumption 1. This is because the ordering of θ_k can be changed arbitrarily by changing y_f . Please clarify this in the paper.**

We are not completely sure about the intent of the reviewer on this point and so we will explain ourselves more clearly here. The Np eigenvalues of the composite output controllability Gramian are computed in descending order, $\mu_0 \geq \mu_1 \geq \dots \geq \mu_{Np-1}$. The associated eigenvectors are also computed, $\xi_k, k = 0, 1, \dots, Np - 1$. The scalar values θ_k^2 are,

$$\theta_k^2 = \left(\beta^T \xi_k \right)^2 \tag{5}$$

where $\beta = [\beta_0^T, \dots, \beta_{N-1}^T]^T$ consists of the control maneuvers stacked on top of each other, $\beta_j = Ce^{A_j t_j} x_0 - y_f$. While changing the vector y_f may change the specific values of θ_k^2 , and may even cause some pairs of values such that $\theta_k^2 > \theta_{k+1}^2$, the over-all behavior decays exponentially still for all cases we examined. Also, the values of θ_k^2 cannot be changed arbitrarily, as there are N copies of $y_f \in \mathbb{R}^p$ stacked on top of each other when constructing β . While the specific values of θ_k^2 may be changed, the exponential scaling overall still holds regardless of the values in y_f .

7. The matrix B is defined in two inconsistent ways in Example 1. One is for what shown in the main text and one is more general for the supplementary.

The reviewer is correct. Our results hold for any choice of the matrix B , as explained in the SI, where B is specified in the main text to be a matrix having elementary vectors as columns. We have added a sentence to clarify that the particular choice of the matrix B presented in the main text is done without loss of generality.

8. What is the point of assuming that the parameter supports be finite? I don't see any use of it in the paper.

We thank the reviewer for advising us to re-visit this point. We have since demonstrated that our assumptions hold for at least one situation where the interval does not have finite support and so we have removed this requirement.

9. In the supplementary, the sentence following (S8) is incorrect. There is no optimization problem remaining. There is only a set of equations to solve to find $x(t_f)$. That what the authors do as well, solve for $x(t_f)$ from the constraints, no free variable is left after applying all the conditions of maximum principle.

We thank the reviewer for pointing this out. That sentence has been rewritten following the reviewer's recommendation.

**REFERENCES**

- ¹M. Beguerisse-Díaz, G. Bosque, D. Oyarzún, J. Picó, and M. Barahona, “Flux-dependent graphs
for metabolic networks,” *npj Systems Biology and Applications* **4**, 32 (2018).
- ²B. D. Anderson and J. B. Moore, *Optimal control: linear quadratic methods* (Courier Corpora-
tion, 2007).
- ³M. A. Dahleh and I. J. Diaz-Bobillo, *Control of uncertain systems: a linear programming ap-
proach* (Prentice-Hall, Inc., 1994).
- ⁴K. Zhou, J. C. Doyle, K. Glover, *et al.*, *Robust and optimal control*, Vol. 40 (Prentice hall New
Jersey, 1996).
- ⁵H. Kimura, Y. Lu, and R. Kawatani, “On the structure of h infinity control systems and related
extensions,” *IEEE Transactions on Automatic Control* **36**, 653–667 (1991).
- ⁶A. A. Stoorvogel, “The h infinity control problem: a state space approach,” Department of Elec-
trical Engineering and Computer Science University of Michigan Ann Arbor USA (1992).
- ⁷R. Freeman and P. V. Kokotovic, *Robust nonlinear control design: state-space and Lyapunov
techniques* (Springer Science & Business Media, 2008).
- ⁸M. Vidyasagar, “Control system synthesis: a factorization approach, part ii,” *Synthesis lectures
on control and mechatronics* **2**, 1–227 (2011).

Reviewer #1 (Remarks to the Author):

The authors have improved the presentation of the work, answering satisfactorily to my comments.

I think that the paper is now suitable for publication in Nature Communications.

Just as one historical note that might not be clear to people not strictly in the field, the first paper that has classified networks ensembles as canonical and microcanonical is Anand & Bianconi PRE 2009 and the non equivalence of the ensembles was already discussed in this paper and fully demonstrated for the first time in Anand Bianconi PRE 2010 despite claims this for found many years later by other authors.

Reviewer #2 (Remarks to the Author):

The authors have made a decent effort to improve the paper and address my concerns, but unfortunately my main concern regarding the significance of the contributions remains. In the rebuttal letter, in response to my main criticism, a number of points are mentioned:

1) "The case the author is presenting ... the control energy.": I think the authors have not understood what I meant by setting $u = 0$. I did not mean a case in which $u = 0$ is the optimal solution to Eq. (4). Instead, I was pointing to the fact that $u = 0$ (or $u = 1$, or $u = 8.3434$, whatever) is a feasible value, and it gives a cost value $J(u = 0)$ which is by definition larger than the optimal cost $J(u = u^*)$. But even $J(u = 0)$ is finite, and so is $J(u = u^*)$. Anyhow, this is not a point of disagreement.

2) "The reviewer is correct that ... additional requirements on b.": I understand the refocusing of the paper and the authors' point, but I think boundedness of optimal control energy is more or less a direct consequence of the boundedness of average deviation. Here is why. Let $\alpha = Np/(Np+b)$, as chosen in the paper. For $N \rightarrow \infty$, this makes $2*J$ equal to the control energy plus sample average of output deviation. Again, setting $u = 0$ (or any other value, as a feasible point) gives a bounded J , say $J\text{-bar}$, because the second term is bounded. Now if we optimize over u , the resulting u^* cannot be (or converge to) infinity because it has to give $J^* \leq J\text{-bar} < \infty$.

The situation with constant α is different, and in fact interesting. First, please note that the control energy does not need to diverge to infinity if α is constant. Only if the lower bound on E goes to infinity, we can say that E goes to infinity as well. And the limit of $1/(Np) * (1/r_1)^{\log((1-\alpha)Np/\alpha)}$ as N goes to infinity may be 0, if $r_1 > 1/e$, or infinity, if $r_1 < 1/e$.

Now assume, as is that case in Example 1, that $r_1 < 1/e$ and the control energy corresponding to constant α diverges to infinity. This is a non-trivial and interesting observation, but it is not well developed (or even clearly mentioned) in the paper. The reason I think it is interesting is because with constant α , the contribution of E in J goes to 0 as N increases. So control energy becomes cheaper and cheaper. In the limit, control energy is free. But even with free control energy, it is not trivial to me that minimizing the expected value of output deviation may end up using infinite energy. So the contrast between the $r_1 > 1/e$ case and the $r_1 < 1/e$ case is very interesting, and something that the authors can potentially build on.

3) "To the second point ... in the manuscript.": Unless I am missing the authors' point, the provided argument is clearly false. If a function is monotonically decreasing and positive, it need not converge to 0. $1 + 1/b$ is such a function, so is $1000 + 1/b$, and none of them converge to 0. In fact, the value of the upper bound as $b \rightarrow \infty$ can remain arbitrarily large, which goes back to my point last time: we only know a bound exists, which is expected, and its value depends on extra variables r_1, \dots from Assumptions 1 & 2 which cannot be computed except using simulations (in which case, the actual value of E and J can be computed using simulations as well). So how can this be useful in any real application?

Also, please remove the related claims in the main article, such as " Thus, there exists a value of b such that the average deviation can be made smaller than any positive value ϵ ." on line 200.

**Controlling network ensembles**

**Response to Reviewers**

Isaac Klickstein^{1, a)} and Francesco Sorrentino^{1, b)}

*Department of Mechanical Engineering, University of New Mexico, Albuquerque,*

*NM 87131*

^{a)}Electronic mail: iklick@unm.edu

^{b)}Electronic mail: fsorrent@unm.edu

**I. REVIEWER 1**

**The authors have improved the presentation of the work, answering satisfactorily to my**
**comments. I think that the paper is now suitable for publication in Nature Communications.**
**Just as one historical note that might not be clear to people not strictly in the field, the first**
**paper that has classified networks ensembles as canonical and microcanonical is Anand &**
**Bianconi PRE 2009 and the non equivalence of the ensembles was already discussed in this**
**paper and fully demonstrated for the first time in Anand Bianconi PRE 2010 despite claims**
**this for found many years later by other authors.**

We thank the reviewer for accepting our responses. We have included these additional refer-
ences to the original definitions of the canonical and microcanonical ensembles as applied to
networks for our discussion therein to be thorough with respect to the previous literature.

**II. REVIEWER 2**

**The authors have made a decent effort to improve the paper and address my concerns,**
 **but unfortunately my main concern regarding the significance of the contributions remains.**
 **In the rebuttal letter, in response to my main criticism, a number of points are mentioned:**

We hope that our responses to the reviewers' comments will convince the reviewer that our
 contributions are significant.

**"The case the author is presenting ... the control energy.": I think the authors have not**
 **understood what I meant by setting $u = 0$. I did not mean a case in which $u = 0$ is the optimal**
 **solution to Eq. (4). Instead, I was pointing to the fact that $u = 0$ (or $u = 1$, or $u = 8.3434$,**
 **whatever) is a feasible value, and it gives a cost value $J(u = 0)$ which is by definition larger**
 **than the optimal cost $J(u = u^*)$. But even $J(u = 0)$ is finite, and so is $J(u = u^*)$. Anyhow, this**
 **is not a point of disagreement.**

We agree with the reviewer that this argument can be shown to demonstrate the control energy
 remains finite given a proper choice of $\alpha(N)$ in a more straightforward manner than we had used.
 The argument can be formally stated as follows. Let $\bar{u}(t)$ be any integrable sub-optimal control
 input, and let γ^{\max} be the resulting maximum difference of any state realization,

$$37 \quad pd^{\max} = \max_{A^{(k)} \in \mathcal{A}} \left\| \left\| y_f - C \left(e^{A^{(k)}(t_f - t_0)} x_0 + \int_{t_0}^{t_f} e^{A^{(k)}(t_f - \tau)} B \bar{u}(\tau) d\tau \right) \right\|_2 \right\|_2^2 \quad (1)$$

The optimal cost can now be upper bounded by,

$$39 \quad J \leq \bar{J} = \frac{1 - \alpha}{2} N d^{\max} + \frac{\alpha}{2} \int_{t_0}^{t_f} \|\bar{u}(t)\|_2^2 dt \quad (2)$$

Note that the energy term is a constant independent of N while the first term grows linearly with
 N . To compensate for this growth, we need $(1 - \alpha)$ to decay as $\frac{1}{N}$, so we choose

$$42 \quad \alpha = \frac{Np}{Np + b} \quad (3)$$

This upper bound can then be expressed as,

$$44 \quad \bar{J} = \frac{b}{2(Np + b)} N p d^{\max} + \frac{Np}{2(Np + b)} \int_{t_0}^{t_f} \|\bar{u}(t)\|_2^2 dt \quad (4)$$

Taking the limit of this expression for constant b yields the asymptotic behavior,

$$46 \quad \lim_{N \rightarrow \infty} \bar{J} = \frac{bd^{\max}}{2} + \frac{1}{2} \int_{t_0}^{t_f} \|\bar{u}(t)\|_2^2 dt \quad (5)$$

which is a constant.

**"The reviewer is correct that ... additional requirements on b .":** I understand the refo-
**ocusing of the paper and the authors' point, but I think boundedness of optimal control**
**energy is more or less a direct consequence of the boundedness of average deviation. Here is**
**why. Let $\alpha = \frac{Np}{(Np+b)}$, as chosen in the paper. For $N \rightarrow \infty$, this makes $2J$ equal to the control**
**energy plus sample average of output deviation. Again, setting $u = 0$ (or any other value,**
**as a feasible point) gives a bounded J , say \bar{J} , because the second term is bounded. Now if**
**we optimize over u , the resulting u^* cannot be (or converge to) infinity because it has to give**
**$J^* \leq \bar{J} < \infty$.**

We agree with the reviewer. By following the reviewer's argument and our derivations pre-
sented above, we can prove that J is upper bounded given a proper choice of the parameter $\alpha(N)$.
Based on the reviewer's feedback, we have removed claims throughout the manuscript that we can
show that the control energy remains finite in the limit of infinitely many realizations. Instead, the
revised manuscript focuses on showing how 'good' or 'bad' formulations of the objective function
can make the optimal control problem either feasible or infeasible in the large N limit. We also
present our derivations above to show boundedness of the optimal control energy. We hope the
reviewer will find our revised presentation of the paper and our added discussion satisfactory.

**The situation with constant α is different, and in fact interesting. First, please note that**
**the control energy does not need to diverge to infinity if α is constant. Only if the lower**
**bound on E goes to infinity, we can say that E goes to infinity as well. And the limit of**
**$\frac{1}{Np} \left(\frac{1}{r_1}\right)^{\log\left(\frac{(1-\alpha)Np}{\alpha}\right)}$ as N goes to infinity may be 0, if $r_1 > 1/e$, or infinity, if $r_1 < 1/e$.**

**Now assume, as is that case in Example 1, that $r_1 < 1/e$ and the control energy corre-**
**sponding to constant α diverges to infinity. This is a non-trivial and interesting observation,**
**but it is not well developed (or even clearly mentioned) in the paper. The reason I think it is**
**interesting is because with constant α , the contribution of E in J goes to 0 as N increases. So**

**control energy becomes cheaper and cheaper. In the limit, control energy is free. But even**
**with free control energy, it is not trivial to me that minimizing the expected value of output**
**deviation may end up using infinite energy. So the contrast between the $r_1 > 1/e$ case and**
**the $r_1 < 1/e$ case is very interesting, and something that the authors can potentially build on.**

We thank the reviewer for proposing an interesting extension of our work. As the reviewer
can easily verify, our assumption 1 is that $\mu_k \approx \mu_0 r_1^k$, where the eigenvalues are ordered such
that $\mu_k \geq \mu_{k+1}$. The approximation implies that $0 < r_1 < 1$ so that $\frac{1}{r_1} > 1$. Then the limit of
$1/(Np) * (1/r_1)^{\log((1-\alpha)Np/\alpha)}$ as N goes to infinity is infinity, independent of the particular value
of $r_1 \in (0, 1)$.

**"To the second point ... in the manuscript.": Unless I am missing the authors' point, the**
**provided argument is clearly false. If a function is monotonically decreasing and positive,**
**it need not converge to 0. $1 + 1/b$ is such a function, so is $1000 + 1/b$, and none of them**
**converge to 0. In fact, the value of the upper bound as $b \rightarrow \infty$ can remain arbitrarily large,**
**which goes back to my point last time: we only know a bound exists, which is expected, and**
**its value depends on extra variables r_1, \dots from Assumptions 1 & 2 which cannot be com-**
**puted except using simulations (in which case, the actual value of E and J can be computed**
**using simulations as well). So how can this be useful in any real application?**

The reviewer is correct that the fact that a function is monotonically decreasing and positive,
does not imply it converges to 0 as the argument approaches infinity. However, this is not our
argument, rather, that we have a function $\bar{D}_N(b)/Np = f(b)$ defined on the interval $b \in [0, \infty)$ with
the properties

1. $f(0) > 0$ (using Eq. (S44))

2. $\lim_{b \rightarrow 0} f(b) = 0$ (using Eqs. (S44) and (S45))

3. $\frac{df}{db} < 0$ (using Eq. (S47))

which imply that $f(b)$ is monotonically decreasing from some positive constant to zero. Two
further statements should be made:

• Finding the positive constant, $\frac{\bar{D}_N(0)}{Np}$, does not require knowing any of the parameters from

Assumptions 1 & 2 as $b = 0$ corresponds to the $\mathbf{u}^*(0) = 0$ case, i.e., the free evolution of
every system.

- • As the average deviation is monotonically decreasing on the interval $\bar{D}(b)/Np \in (0, \bar{D}_N(0)/Np]$,
then there exists an inverse function that provides the value of b for any desired average de-
viation. This is the use in applications where we can guarantee that for any demanded
average deviation, one can find a value of b to satisfy that demand.

**Also, please remove the related claims in the main article, such as " Thus, there exists a**
**value of b such that the average deviation can be made smaller than any positive value ep-**
**silon." on line 200.**

Please see our response above.

Reviewer #2 (Remarks to the Author):

I want to thank the authors for their response, but unfortunately not much of my concerns from the last round are properly addressed. I bring below what is still outstanding.

- Re: "We agree with the reviewer. By following ... discussion satisfactory."

The authors say that "the revised manuscript focuses on showing how 'good' or 'bad' formulations of the objective function can make the optimal control problem either feasible or infeasible in the large N limit.", but to my understanding this has been the focus of the paper from the beginning. It was maybe worded a bit differently. If by 'bad' formulation the authors mean a constant alpha and by a 'good' formulation they mean what they propose, I still think this is near-trivial. It is essentially nothing but dividing a sum over N terms by N to make it an average before adding it up with control energy, which I don't see why the authors did not even do so in the first place. In other words, by defining the objective function as a convex combination in Eq. (4), the authors are creating a problem which then they try to solve using convoluted arguments, while the natural choice of such an objective function could have been a division by N for the first term.

- Re: "We thank the reviewer for proposing an interesting ... (0,1)."

This is not true. I'm quite disappointed that the authors did not even check what I told them. Just plug in $r_1=0.5$ or any other $r_1>1/e$ and you can easily see numerically that $(1/r_1)^{\log(N)}/N$ converges to zero.

- Re: "The reviewer is correct that the fact that a function ... satisfy that demand."

Now I see the authors' argument better. I hadn't understood it correctly last time. I haven't yet checked all the details of the derivations in S1.5, but I assume they are correct. My problem is bigger. Even if the limit of the $Dbar_N(b)/Np$ as b goes to infinity is 0, that is for a fixed N. But the whole focus and motivation of the paper is for N to infinity. At first I thought this is a minor point, but now I think it is a pretty fundamental inconsistency. Note that b is a constant in the denominator of alpha, which can be chosen arbitrarily but has to be chosen and fixed BEFORE N is increased to infinity. And ironically, increasing b and increasing N have completely opposite effects on alpha. N to infinity takes alpha to 1 (which is the whole story of the paper), while b to infinity (for any FIXED N) takes alpha to 0. So you can choose b as large as you want, but that only shifts the initial value of alpha towards 0, which has 2 problems: 1) it places all the weight of the objective function on deviations and makes energy very cheap, which blows up energy as the authors acknowledge, and 2) it becomes irrelevant as N goes to infinity, which is the thermodynamic limit the entire paper is about.

**Controlling network ensembles**

**Response to Reviewers**

Isaac Klickstein^{1, a)} and Francesco Sorrentino^{1, b)}

*Department of Mechanical Engineering, University of New Mexico, Albuquerque,*

*NM 87131*

^{a)}Electronic mail: iklick@unm.edu

^{b)}Electronic mail: fsorrent@unm.edu

I. REVIEWER 2

I want to thank the authors for their response, but unfortunately not much of my con-
cerns from the last round are properly addressed. I bring below what is still outstanding.

We provide detailed responses below.

The authors say that "the revised manuscript focuses on showing how 'good' or 'bad'
formulations of the objective function can make the optimal control problem either feasible
or infeasible in the large N limit.", but to my understanding this has been the focus of the
paper from the beginning. It was maybe worded a bit differently. If by 'bad' formulation
the authors mean a constant α and by a 'good' formulation they mean what they propose,
I still think this is near-trivial. It is essentially nothing but dividing a sum over N terms
by N to make it an average before adding it up with control energy, which I don't see why
the authors did not even do so in the first place. In other words, by defining the objective
function as a convex combination in Eq. (4), the authors are creating a problem which then
they try to solve using convoluted arguments, while the natural choice of such an objective
function could have been a division by N for the first term.

By 'good' formulation, we meant one in which we parameterize the weighting coefficient in
such a way that both the control energy is bounded and there exists a parameter to arbitrarily ad-
just this upper bound. The reviewer suggests we choose $\alpha = \frac{Np-1}{Np}$ (which makes $(1 - \alpha) = \frac{1}{Np}$),
or in other words dividing the first term by N . While the reviewer is correct that this choice bounds
the control energy, it does not introduce a parameter capable of adjusting the upper bound of the
control energy. One could then attempt to introduce a parameter by defining $\alpha = \frac{Np-b_2}{Np}$ but note
that b_2 is restricted to the interval $b_2 \in (0, Np)$ to ensure $\alpha \in (0, 1)$. This means that b_2 cannot be
adjusted independently of the size of the system, and in fact if we attempt to relate this alternative
parametrization to the original one, $\alpha = \frac{Np}{Np+b}$, we find that for any choice of b , the corresponding
value of b_2 is,

$$34 \quad b_2 = \frac{Npb}{Np+b} \quad (1)$$

The importance of this relation arises from the fact we have already shown that the upper bound
can be arbitrarily reduced or raised for some value of b , but the corresponding value of b_2 depends

on the system size.

Another option would be $\alpha = \left(\frac{Np-1}{Np}\right)^{b_3}$. In this case, we the parameter b_3 must be restricted to,

$$39 \quad \frac{1}{\log(Np-1) - \log(Np)} < b_3 < \infty \quad (2)$$

to ensure $\alpha \in (0, 1)$. While this choice of parameterization may be equally capable of bounding
the control energy, the resulting form is not as clear as what we derived with our choice of $\alpha(N)$,

$$42 \quad \bar{E}_N(b) \leq b^2 K \quad (3)$$

which makes the relation between the parameter and the control energy very clear.

The reviewer may still believe that our choice of $\alpha(N)$ is a trivial but our choice is the one capable
of both arbitrarily bounding the control energy and leads to clear, simple, bounds on the control
energy unlike the choices $\alpha = const$, $\alpha = \frac{Np-b_2}{Np}$, or $\alpha = \left(\frac{Np-1}{Np}\right)^{b_3}$.

**This is not true. I'm quite disappointed that the authors did not even check what I told**
**them. Just plug in $r_1 = 0.5$ or any other $r_1 > 1/e$ and you can easily see numerically that**
**$(1/r_1)^{\log(N)/N}$ converges to zero.**

We have derived a far simpler lower bound for the control energy in the revised supplement-
tary information that better encompasses the points we are trying to make. We believe that there
was a notation mistake on our part that lead to the confusion. We apologize for this oversight.

For clarity, the new bounds for the energy approximation that appear in the SI are,

$$56 \quad c_1 c_2 \left(\frac{b}{1+c_1 b}\right)^2 \left(\frac{1-(r_1 r_2)^{Np}}{1-r_1 r_2}\right) \leq \bar{E}_N(b) \leq b^2 \left[c_1 c_2 \left(\frac{1-(r_1 r_2)^{\bar{k}+1}}{1-r_1 r_2}\right) + \frac{c_1 \theta_c^2}{Np} \left(\frac{r_1^{\bar{k}+1} - r_1^{Np}}{1-r_1}\right) \right] \quad (4)$$

which has eliminated the appearance of the equation in question.

**Now I see the authors' argument better. I hadn't understood it correctly last time. I haven't**
**yet checked all the details of the derivations in S1.5, but I assume they are correct. My prob-**
**lem is bigger. Even if the limit of the $\bar{D}_N(b)/Np$ as b goes to infinity is 0, that is for a fixed**
**N . But the whole focus and motivation of the paper is for N to infinity. At first I thought this**
**is a minor point, but now I think it is a pretty fundamental inconsistency. Note that b is a**
**constant in the denominator of alpha, which can be chosen arbitrarily but has to be chosen**

**and fixed BEFORE N is increased to infinity. And ironically, increasing b and increasing N**
**have completely opposite effects on α . N to infinity takes α to 1 (which is the whole story of**
**the paper), while b to infinity (for any FIXED N) takes α to 0. So you can choose b as large**
**as you want, but that only shifts the initial value of α towards 0, which has 2 problems: 1)**
**it places all the weight of the objective function on deviations and makes energy very cheap,**
**which blows up energy as the authors acknowledge, and 2) it becomes irrelevant as N goes**
**to infinity, which is the thermodynamic limit the entire paper is about.**

We once again thank the reviewer for thinking carefully about our derivations. We agree with
the reviewer that we want to show that for a fixed value of b the control energy is bounded for any
finite N and even in the thermodynamic limit. This is exactly what we do in the paper: from Eq.
(11) we see that the limit $\lim_{N \rightarrow \infty} \bar{J}_N(b)$ is a constant independent of N . In addition to that, we
also took limits in b to show that there exists a value of b for which the upper bound on the control
energy could be made arbitrarily small. To further clarify this point, we have slightly rewritten the
discussion following Eq. (11) of the main manuscript.